PCSK9 inhibitor effectively alleviated cognitive dysfunction in a type 2 diabetes mellitus rat model

Yang Yang 1
Wang Yeying 2
Wang Yuwen 1
Ke Tingyu 1 ketingyu@kmmu.edu.cn
Zhao Ling 1 zhaoling@kmmu.edu.cn
1 Department of Endocrinology, the Second Affiliated Hospital, Kunming Medical University , Kunming, Yunnan , China
2 Department of Epidemiology and Biostatistics, School of Public Health, Kunming Medical University , Kunming, Yunnan , China
Gould Gwyn
Electronic publication date: 2024 Aug 14
Publication date: 2024
Volume: 12
Electronic Location ID: e17676
Received 2023 Oct 30; Accepted 2024 Jun 12
Copyright: © 2024 Yang et al.
Copyright year: 2024
Copyright holder: Yang et al.
License: This is an open access article distributed under the terms of the Creative Commons Attribution License, which permits unrestricted use, distribution, reproduction and adaptation in any medium and for any purpose provided that it is properly attributed. For attribution, the original author(s), title, publication source (PeerJ) and either DOI or URL of the article must be cited.
License URL: https://creativecommons.org/licenses/by/4.0/

Keywords: Diabetes-associated cognitive dysfunction, PCSK9 inhibitors, Proteomics, Parallel reaction monitoring

Funding: Yunnan Provincial Health Commission Medical Leading Talent Training Program L-2019014 Joint Special Project of Applied Basic Research of Kunming Medical University Department of Science and Technology of Yunnan Province 202201AY070001-127 and 202201AT070242 Yunnan High-level Talent Cultivation Support Programme YNWR-MY-2020-022 National Natural Science Foundation of China-Yunnan Joint Fund 82201475 The research was supported by the Yunnan Provincial Health Commission Medical Leading Talent Training Program (Grant no. L-2019014), the Joint Special Project of Applied Basic Research of Kunming Medical University, Department of Science and Technology of Yunnan Province (202201AY070001-127; 202201AT070242), the Yunnan High-level Talent Cultivation Support Programme (YNWR-MY-2020-022), and the National Natural Science Foundation of China-Yunnan Joint Fund (82201475). The funders had no role in study design, data collection and analysis, decision to publish, or preparation of the manuscript.

==============================
Background

The incidence of diabetes-associated cognitive dysfunction (DACD) is increasing; however, few clinical intervention measures are available for the prevention and treatment of this disease. Research has shown that proprotein convertase subtilisin/kexin type 9 (PCSK9) inhibitors, particularly SBC-115076, have a protective effect against various neurodegenerative diseases. However, their role in DACD remains unknown. In this study, we aimed to explore the impact of PCSK9 inhibitors on DACD.

Methods

Male Sprague-Dawley (SD) rats were used to establish an animal model of type 2 diabetes mellitus (T2DM). The rats were randomly divided into three groups: the Control group (Control, healthy rats, n = 8), the Model group (Model, rats with T2DM, n = 8), and the PCSK9 inhibitor-treated group (Treat, T2DM rats treated with PCSK9 inhibitors, n = 8). To assess the spatial learning and memory of the rats in each group, the Morris water maze (MWM) test was conducted. Hematoxylin-eosin staining and Nissl staining procedures were performed to assess the structural characteristics and functional status of the neurons of rats from each group. Transmission electron microscopy was used to examine the morphology and structure of the hippocampal neurons. Determine serum PCSK9 and lipid metabolism indicators in each group of rats. Use qRT-PCR to detect the expression levels of interleukin (IL)-1β, IL-6, and tumor necrosis factor-alpha (TNF-α) in the hippocampal tissues of each group of rats. Western blot was used to detect the expression of PCSK9 and low-density lipoprotein receptor (LDLR) in the hippocampal tissues of rats. In addition, a 4D label-free quantitative proteomics approach was used to analyse protein expression in rat hippocampal tissues. The expression of selected proteins in hippocampal tissues was verified by parallel reaction monitoring (PRM) and immunohistochemistry (IHC).

Results

The results showed that the PCSK9 inhibitor alleviated cognitive dysfunction in T2DM rats. PCSK9 inhibitors can reduce PCSK9, total cholesterol (TC), and low-density lipoprotein (LDL) levels in the serum of T2DM rats. Meanwhile, it was found that PCSK9 inhibitors can reduce the expression of PCSK9, IL-1β, IL-6, and TNF-α in the hippocampal tissues of T2DM rats, while increasing the expression of LDLR. Thirteen potential target proteins for the action of PCSK9 inhibitors on DACD rats were identified. PRM and IHC revealed that PCSK9 inhibitors effectively counteracted the downregulation of transthyretin in DACD rats.

Conclusion

This study uncovered the target proteins and specific mechanisms of PCSK9 inhibitors in DACD, providing an experimental basis for the clinical application of PCSK9 inhibitors for the potential treatment of DACD.

Introduction

Epidemiological data have shown that 60–70% of patients with diabetes mellitus (DM) exhibit mild to moderate cognitive impairment, which significantly increases the risk of dementia and places a heavy burden on the patient’s family and on society (Chornenkyy et al., 2019; Moran et al., 2019). Previous research has indicated that, as the disease progresses, diabetic patients gradually develop cognitive impairments, including decreased learning and memory abilities, as well as diminished spatial cognition (Hassing et al., 2004; Mayeda et al., 2015). Studies suggest that DM can lead to diabetes-associated cognitive dysfunction (DACD), a state that is characterized by cognitive impairment that lies somewhere between normal cognitive function and dementia (Dong et al., 2016; Sadanand, Balachandar & Bharath, 2016). In 2021, the American Diabetes Association’s guidelines clearly identified DACD as a common complication of type 2 diabetes mellitus (T2DM) (American Diabetes Association, 2021). In recent years, DACD has become a common yet frequently overlooked chronic complication of DM in clinical practice (Meng et al., 2017; Prieto-Gomez, Diaz-Vazquez & Perez-Torres, 2020). DACD represents a complication that bridges endocrinology and neurology. In general, DACD is difficult to reverse and may progress to dementia. The incidence of DACD is on the rise, leading to progressively serious medical and social challenges. Therefore, the early detection and treatment of DACD are crucial. Unfortunately, the existing supportive therapies for DACD are largely ineffective. Therefore, efforts to elucidate specific treatments and preventive strategies for DACD remain important.

Proprotein convertase subtilisin/kexin type 9 (PCSK9) is a 692-amino acid serine protease that is primarily expressed in hepatocytes (Horton, Cohen & Hobbs, 2009). The American Food and Drug Administration have approved the use of a monoclonal antibody as a novel lipid-lowering medication that targets PCSK9 (Chaudhary et al., 2017; Fala, 2016; Hess et al., 2018). These monoclonal antibodies inactivate PCSK9, thereby reducing the degradation of LDLR and consequently lowering LDL levels in the blood (Momtazi-Borojeni et al., 2019). Recent research has indicated that PCSK9 inhibitors can mitigate brain pathologies, such as vascular events in stroke (Castilla-Guerra, Fernandez-Moreno & Rico-Corral, 2019), prevent brain injury resulting from myocardial ischemia/reperfusion damage (Apaijai et al., 2019), and reduce insulin resistance-related cognitive impairment (Arunsak et al., 2020). However, as yet, no studies have examined the impact of PCSK9 inhibitors on DACD-related brain functions.

In patients with DACD, levels of TNF-α and IL-6 are elevated in the blood, and these levels are correlated with the severity of cognitive dysfunction (Gorska-Ciebiada et al., 2015; Ragy & Kamal, 2017). Animal experiments also confirm a significant increase in IL-6 and TNF-α levels in the hippocampal tissues of DACD rats (Chen et al., 2022). Several studies have also found that inflammation plays a crucial role in the damage induced by chronic degenerative neurological diseases such as Alzheimer’s disease (AD) and Parkinson’s disease (PD) (Dzamko, Geczy & Halliday, 2015; Verdile et al., 2015). The relationship between diabetic inflammation and cognitive dysfunction has gradually become a hot topic in recent years (Herder et al., 2017). Previous studies have found that PCSK9 inhibitors treatment can reduce the expression of IL-6 in endothelial cells (D’Onofrio et al., 2023), the expression of TNF-α and IL-6 in atherosclerotic tissues (Yang et al., 2023), and the expression of IL-1β and TNF-α in carotid plaque. However, it is still unknown whether PCSK9 inhibitors can affect the inflammatory factors in DACD hippocampal tissues. Therefore, this study conducted Quantitative real-time PCR (qRT-PCR) to detect the levels of inflammatory factors IL-1β, IL-6, and TNF-α in the hippocampal tissues of each group of rats.

Proteins play a crucial role in virtually all cellular functions (Qiao et al., 2018). Proteomics techniques allow for the qualitative and quantitative analysis of hundreds of proteins, revealing comprehensive protein changes. 4D label-free technology is a proteomics technique that identifies differentially expressed proteins (DEPs) (Meier et al., 2018). In this study, 4D label-free quantitative proteomics was used to explore the impact of PCSK9 inhibitor treatment on rats with DACD. This analysis involved the examination of hippocampal tissue samples from three distinct rat groups (healthy rats, T2DM rats, and T2DM rats treated with PCSK9). This research provides new theoretical evidence for the use of PCSK9 inhibitors in the prevention and treatment of DACD.

Materials and Methods

Ethics statement

All experiments and procedures in this study were performed in strict accordance with the protocols approved by the Institutional Animal Care and Use Committee of Kunming Medical University (Approval number: kmmu20211142).

Animal models

A total of 24 healthy male Sprague–Dawley (SD) rats (180–220 g) were obtained from Kunming Medical University (License No: SYXK; Kunming, China; K2020-0006). All rats were housed in a specific-pathogen-free grade animal observation room at Kunming Medical University that was maintained at a temperature of 20–24 °C and a relative humidity of 60%, with a 12-h light/dark cycle. All animals were provided with a standard autoclaved commercial diet and filtered water. Our research model was built after 7 days of adapting the rats’ diets. The rats were randomly divided into three groups: the Control group (Control, healthy rats, n = 8), the Model group (Model, rats with T2DM, n = 8), and the PCSK9 inhibitor group (Treat, T2DM rats treated with PCSK9 inhibitors, n = 8). The Control group was maintained on a regular diet, while the other rats were fed a HFD for 6 weeks to induce T2DM. Following 6 weeks of HFD feeding, the rats underwent a 12-h fast before receiving an intraperitoneal injection of streptozotocin (STZ; Sigma, Burlington, MA, USA) at a dose of 35 mg/kg body weight (Chen et al., 2021). The control rats were administered an equivalent volume of sodium citrate buffer. After the STZ injection, rats in the Treat and Model groups were maintained on the HFD, while the Control group received a regular diet. Rats with fasting blood glucose levels exceeding 16.7 mmol/L after STZ injection were diagnosed with T2DM (Chen et al., 2021). All Model rats had blood glucose levels greater than 16.7 mmol/L after STZ injection. After a stable model period of 4 weeks, the Treat group received subcutaneous injections of the PCSK9 inhibitor, SBC-115076, at a dose of 4 mg/kg/day (SBC-115076; Selleck Chemicals, Texas, USA) for 4 weeks, while the Control and Model groups received subcutaneous injections of an equal volume of double-distilled water for the same duration.

Animal euthanasia

Since it was necessary to collect hippocampal tissue in this study, the rats were euthanized at the end of the experimental period. All rats were sacrificed using an intraperitoneal injection of 1% pentobarbital sodium (40 mg/kg). Record the body weight of each group of rats before euthanization.

Morris water maze test

After 4 weeks of PCSK9 inhibitor treatment, the rats were subjected to 6 days of trial in the Morris water maze (MWM) (Morris et al., 2006) to investigate their spatial learning ability and memory function. This experiment included eight rats from each group, making a total of 24 rats. The experimental setup consisted of a circular pool (diameter, 120 cm; height, 60 cm), filled with opaque water (depth 30 cm, temperature 24–26 °C), which was evenly divided into four quadrants, labeled northwest, northeast, southwest, and southeast. A transparent escape platform was placed 2 cm below the water surface in the southwest quadrant (target quadrant). The test consisted of two test phases: the place navigation test and the spatial probe test. Rats underwent continuous training for 5 days for the place navigation test, with each trial lasting 90 s, and the escape latency for each trial was recorded. In these trials, rats were released into the water, facing the pool wall, from four different starting positions around the pool. If a rat found the platform within 90 s, it was allowed to stay on it for 3 s. If a rat failed to find the platform within 90 s, it was manually guided to the platform and allowed to rest there for 15 s. For the spatial probe tests, the platform was removed. Then a rat was released from the quadrant of the pool farthest from the primary platform, and the time the rats spent in the target quadrant was recorded, as well as the number of times they crossed the platform area within a 90-s period.

Specimen collection and storage

After the MWM test, rats were anesthetized and euthanized using an intraperitoneal injection of 1% pentobarbital sodium (40 mg/kg). The hippocampi were removed and collected from both hemispheres. Some samples were separated and fixed in 4% paraformaldehyde for subsequent staining with hematoxylin and eosin (H&E) and Nissl staining. Hippocampal cornu ammonis 1 (CA1) regions of the brain were dissected into 1 mm3 samples and fixed in 2.5% glutaraldehyde at 4 °C for transmission electron microscope (TEM) analysis. The remaining samples were collected and stored at −80 °C for proteomics and molecular experiments.

H&E and Nissl staining

This experiment involved three rats, one from each group. Hippocampal tissues fixed for more than 24 h were sequentially cleaned and dehydrated in a dehydration apparatus containing gradient alcohol. Subsequently, the dehydrated hippocampal tissues were embedded in paraffin and sectioned into 3 μm thick slices using a paraffin microtome (RM2235; Leica Biosystems, Wetzlar, Germany). H&E staining was employed to assess pathological changes in the hippocampal CA1 region, while Nissl staining, using 0.1% cresyl violet, was used to analyze the functional changes in neurons. Images were observed and analyzed under a microscope at a magnification of 400X (Eclipse E100, Nikon, Japan).

Transmission electron microscopy

One rat from each group was selected for this experiment. Several blocks of 2.5% glutaraldehyde-fixed hippocampal tissue were rinsed, dehydrated, embedded, and cured. Subsequently, ultrathin sections (<60 nm) were cut using an ultramicrotome (Leica Microsystems, Wetzlar, Germany) and stained with a 2% uranyl acetate-saturated alcohol solution for 15 min. After rinsing with double-distilled water, the sections were stained with lead citrate solution for 15 min. After drying, each section was examined under transmission electron microscopy (TEM) (HT7700; Hitachi, Tokyo, Japan) to observe the ultrastructure within the neurons.

Measurement of serum PCSK9, glucose and lipid metabolism indicators

Serum samples were collected, and serum FBG, TC, triglyceride (TG), LDL and high-density lipoprotein (HDL) levels were measured with an automated biochemical analyser (Siemens, Munich, Germany). The serum fasting insulin (FINS) and PCSK9 levels were measured using rat ELISA kits (Quanzhou Jiubang Biotechnology Co., Ltd, Guangzhou, China).

Quantitative real-time polymerase chain reaction

We isolated total RNA from fresh rat hippocampal tissue using the TRIZOL. Subsequently, 1 μg of total RNA was reverse transcribed into cDNA using the GoScript™ Reverse Transcription System. Quantitative real-time polymerase chain reaction (qRT-PCR) was performed on the ABI7500 system using the GoTaq® qPCR Master Mix. Expression levels were normalized to β-actin expression, and relative RNA expression was calculate using the 2−ΔΔCT. Detailed primer information is provided in Table S1.

Protein extraction

Hippocampal tissue samples were ground into cellular powder and then treated with a lysis buffer mixture consisting of four volumes of 8 M urea and 1% protease inhibitor cocktail. The mixture was processed using a high-intensity ultrasonicator. Subsequently, centrifugation was performed at 12,000 × g for 10 min at 4 °C, and the supernatant was collected. The protein concentration was determined using a BCA kit (Abcam; Abcam PLC, Cambridge, UK).

Western blot analysis

After separating the proteins using SDS-PAGE, they were transferred onto PVDF membranes. Subsequently, the membranes were blocked with 5% nonfat milk. After washing three times with TBST, the membranes were incubated with primary antibodies against PCSK9 (1:1,500, bioss bs-6060R), LDLR (1:2,000, bioss bs-0705R), and β-actin (GB12001; Wuhan Servicebio Technology Co. LTD., Wuhan, China, diluted to 1:3000) for 1 h at room temperature. Antibodies were purchased from Beijing Biosynthesis Biotechnology Co., LTD. (Beijing, China). After washing three times with TBST, the membranes were incubated with secondary antibodies (1:7,000) at room temperature for 1 h. Subsequently, the immunoblots were visualized using an ECL kit (Roche, Basel, Switzerland), and finally, the band images were quantified using ImageJ software.

Trypsin digestion

Add 5 mM dithiothreitol (Sigma, Burlington, MA, USA) to the protein solution and incubate at 56 °C for 30 min. Then, add 11 mM iodoacetamide (Sigma, Burlington, MA, USA) in the dark and incubate at room temperature for 15 min. Next, dilute the protein samples to 100 mM TEAB (Sigma, Burlington, MA, USA). Finally, add trypsin (Promega, Madison, WI, USA) at a ratio of 1:50 (trypsin-to-protein mass ratio) to the solution for overnight digestion, followed by a second digestion at a ratio of 4:1 for 100 h.

Liquid chromatography-tandem mass spectrometry analysis, 4D mass spectrometry

The peptides, dissolved in liquid chromatography mobile phase A (an aqueous solution containing 0.1% formic acid and 2% acetonitrile), were separated using a NanoElute UPLC system. The gradient was set as follows: 0–70 min, 6–24% B; 70–84 min, 24–35% B; 84–87 min, 35–80% B; 87–90 min, 80% B. The flow rate was maintained at 450 nL/min. Mobile phase B contained 0.1% formic acid and 100% acetonitrile. Peptides separated in the UHPLC system were introduced into the capillary ion source for ionization, followed by analysis using the timsTOF Pro mass spectrometer. The ion source voltage was set at 1.65 kV, and high-resolution TOF detection was used to detect and analyze both the precursor ions and their fragment ions. The secondary mass spectrometry scan range was set to 100–1,700, and the data acquisition mode was set to parallel accumulation-serial fragmentation (PASEF) mode. Following each primary mass spectrometry acquisition, ten rounds of secondary spectrum acquisitions were performed in PASEF mode, with precursor ion charge states ranging from zero to five. The dynamic exclusion time for the tandem mass spectrometry scans was set to 30 s to avoid redundant precursor ion scanning.

Bioinformatics analysis

To compare the relative protein expression levels among the three groups, the fold changes (FCs) were calculated for each pair of groups. Proteins with FC values greater than 1.2 or less than 0.83, and a Q-value below 0.05, were identified as DEPs (Sa et al., 2023). Hierarchical clustering analysis of the DEPs was performed using a free online bioinformatics platform (www.bioinformatics.com.cn). Volcano plots and heatmaps were utilized as visual aids using the same platform. The subcellular localization and domain features of the DEPs were analyzed using the CELLO database (http://cello.life.nctu.edu.tw/). The DEPs were subjected to GO annotation and KEGG pathway enrichment analysis using the DAVID tool (https://david.ncifcrf.gov/). Finally, a bubble chart was generated using a free online bioinformatics platform (www.bioinformatics.com.cn) for analysis of the bioinformatics-related data.

Parallel reaction monitoring for target protein validation

To further validate the expression of the DEPs, we performed PRM analysis using the cryopreserved hippocampal samples from the remaining rats (three rats per group). This analysis aimed to quantitatively assess the expression levels of the selected proteins. The methods used for the protein extraction and trypsin digestion were the same as those used in the proteomic analysis experiment, described above. After extraction and trypsin digestion, the peptide samples were analyzed using the TripleTOF 5600 + mass spectrometer in conjunction with the Eksigent microLC system (AB SCIEX, Framingham, MA, USA). The resulting MS/MS data were processed using the ProteinpilotTM V4.5 search engine, followed by additional analysis with the Skyline software. This integrated approach enabled us to generate PRM spectrum files and obtain quantitative information about the proteins.

Immunohistochemistry

The hippocampal tissue paraffin was sliced into 3 μm thick sections. After dewaxing, the sections were incubated overnight at 4 °C with Ttr antibody (bs-0152R, diluted 1:100). Subsequently, the sections were sequentially incubated with secondary antibodies for 15 min, followed by 15 min of peroxidase-conjugated avidin-biotin complex (ABC) incubation, 10 min of 3, 3′-diaminobenzidine (DAB) substrate reaction, and 5 min of counterstaining with hematoxylin. The Integral Optical Density (IOD) and pixel area of positive expression in the tissue sections were analyzed using Image Pro-Plus 6.0 software.

Statistical analysis

Data were expressed as the mean ± standard error of the mean. SPSS 19.0 software (SPSS, Chicago, IL, USA) was used for statistical analysis. For data that followed a normal distribution, ANOVA was employed for comparisons. In the case of the MWM test, repeated measures analysis of variance was applied to analyze the escape latency data. For data that did not follow a normal distribution, nonparametric rank-sum tests were utilized. A p-value of <0.05 was considered statistically significant. The significance levels were indicated as follows: *p < 0.05, **p < 0.01, vs. Control. #p < 0.05, ##p < 0.01, vs. Model.

Results

PCSK9 inhibitors ameliorate learning and memory impairments in T2DM rats

We employed the MWM test to assess the impact of PCSK9 inhibition on spatial learning and memory. The MWM test included the place navigation test and the spatial probe test. The swimming trajectory of the rats on day 5 of the place navigation test is depicted in Fig. 1A. The results showed that the rats in the Model group had a poorer ability to search for the platform and remember its location compared to rats in the Control group. Model rats primarily used edge-based and random-based searching strategies. The rats in the Treat group performed slightly better than those in the Model group. Compared to the Control group, the Model group exhibited a significant increase in escape latency from Day 3 to Day 5 (Fig. 1B). However, compared to the Model group, rats treated with PCSK9 inhibitors exhibited a significant reduction in escape latency from Day 3 to Day 5 (Fig. 1B). To eliminate the potential influence of rat swimming speed on the place navigation test, we measured the swimming speed of the rats. The results showed no significant difference in swimming speed among the three groups of rats (Fig. 1C). The swimming trajectory of the rats during the spatial probe test is illustrated in Fig. 1D. The T2DM rats spent less time in the target quadrant (Fig. 1E) and made fewer platform crossings (Fig. 1F) during the spatial probe test compared to rats in the other groups. However, in comparison to the T2DM group, rats treated with PCSK9 inhibitors spent more time in the target quadrant (Fig. 1E), and exhibited an increased number of platform crossings (Fig. 1F). The results indicate that T2DM rats displayed cognitive deficits, while PCSK9 inhibitor treatment appeared to ameliorate the impaired spatial learning and memory capabilities in T2DM rats.

Figure 1 MWM test of T2DM rat.

(A) Swimming track of T2DM rats on day 5 of the place navigation test; (B) the escape latency during the place navigation test; (C) the swimming speed during the place navigation test; (D) swimming track of T2DM rats during the spatial probe test; (E) time spent in the platform quadrant during the spatial probe test; (F) platform crossing times during the spatial probe test. Data are expressed as mean ± SEM, n = 8. *p < 0.05, **p < 0.01, vs. Control. #p < 0.05, ##p < 0.01, vs. Model.

PCSK9 inhibitors mitigate hippocampal neuronal injuries and ultrastructural deterioration in T2DM rats

Histological analysis of the hippocampal CA1 region of T2DM rats revealed deep nuclear staining and neuronal disarray compared to the Control group (Fig. 2A). In T2DM rats, hippocampal cell death increases (Fig. 2B). In the CA1 region of the hippocampus, Nissl bodies were abundant in the Control group, whereas T2DM rats showed a reduction in Nissl bodies (Figs. 2C, 2D). PCSK9 inhibitor treatment significantly ameliorated these pathological alterations in T2DM rats (Figs. 2B, 2D). Mild cellular swelling was observed in the hippocampal neurons of T2DM rats, as well as a reduced neuron count, blurred, fractured, dissolved mitochondrial cristae, and partial disruption of the rough endoplasmic reticulum in localized regions (Fig. 2E). However, in the treatment group, the neuronal ultrastructure showed varying degrees of improvement (Fig. 2E). Overall, these observations suggest that PCSK9 inhibitors reduce pathological degeneration and enhance the ultrastructure of neurons.

Figure 2 Morphological changes in the hippocampal CA1 area.

(A) The representatives picture of HE staining in hippocampal regions CA1 (200×, 400×) from each group; (B) quantitative statistics of HE staining. (C) Nissl-stained neurons of the hippocampal CA1 (200×, 400×) from each group; (D) quantitative statistics of Nissl staining. (E) Ultrastructural changes in the hippocampus: mitochondria (black arrow), rough endoplasmic reticulum (red arrow) (1,200×, 6,000×), N represents the nucleus of the cell. (N = 3 per group). **p < 0.01, vs. Control. ##p < 0.01, vs. Model.

The effects of PCSK9 inhibitors on body weight, serum PCSK9 levels and lipid metabolism indicators in each group of rats

Compared to the Control group, rats in the Model group showed increases in serum PCSK9, FBG, TC, TG, and LDL levels, as well as decreases in FINS and body weight (Figs. 3A, 3B). Compared to the Model group, rats in the Treat group exhibited decreases in serum PCSK9, TC, and LDL levels (Figs. 3A, 3B). In summary, the results indicate that PCSK9 inhibitors can reduce serum PCSK9, TC, and LDL levels in T2DM rats, with no significant effects on body weight and glycemic metabolism indicators.

Figure 3 Detection of relevant indexes in serum and hippocampal tissues of rats in each group.

(A) Body weight and glycolipid metabolic parameter in each group of rats (N = 8 per group); (B) serum PCSK9 levels of rats in each group (N = 8 per group); (C) the mRNA expression of IL-1β, IL-6, and TNF-α in the hippocampal tissue (N = 5 per group); (D) western blotting analysis of the PCSK9 and LDLR in the hippocampal tissue of each group (N = 3 per group). *p < 0.05, **p < 0.01, vs. Control. #p < 0.05, ##p < 0.01, vs. Model.

Identification of IL-1β, IL-6, and TNF-α mRNA expression in hippocampal tissue by qRT-PCR

The results indicated that the mRNA levels of IL-1β, IL-6, and TNF-α were up-regulated in the Model group compare with the Control group (Fig. 3C). Howeover, compared to the Model group, the mRNA levels of IL-1β, IL-6, and TNF-α were down-regulated in Model group (Fig. 3C).

Identification of PCSK9 and LDLR protein expression in hippocampal tissue by western blot

The western blot results further revealed that PCSK9 expression was elevated in the hippocampal tissue of T2DM rat, and PCSK9 inhibitors effectively inhibited this increase (Fig. 3D). It was also found that LDLR expression decreased in the hippocampal tissue of T2DM rats, and PCSK9 inhibitors effectively inhibited this decrease (Fig. 3D).

4D label-free quantitative proteomics analysis

High-throughput label-free liquid chromatography-tandem mass spectrometry (LC-MS/MS) experiments were employed to analyze the rat hippocampal tissue samples. A total of 5,347 protein species were identified in this study, with 5,307 protein species were quantified (Fig. 4A). Compared to the Control group, the Model group exhibited 59 upregulated and 98 downregulated DEPs, while the Treat group showed 80 upregulated and 188 downregulated DEPs. Compared to the Model group, the Treat group displayed 40 upregulated and 56 downregulated DEPs (Fig. 4B). A heatmap was constructed to illustrate the differences in protein expression between the groups (Fig. 4C).

Figure 4 Analysis of DEPs in three group.

(A) Peptides and proteins. (B) The analysis of DEPs. (C) Hierarchical clustering analysis of DEPs. (N = 3 per group).

Identification of DEPs associated with DACD

To determine which proteins are responsible for DACD, we analyzed the DEPs between the Control and Model groups. A volcano plot was constructed to illustrate the protein differences, revealing 59 upregulated and 98 downregulated DEPs between the Control and Model groups (Fig. 5A). WoLF PSORT was employed to accurately predict the subcellular localization of DEPs, which revealed that the DEPs were primarily located in the nucleus, followed by the cytoplasmic, extracellular, and mitochondrial domains, and the plasma membrane (Fig. 5B). GO terms were categorized into three groups: biological processes (BP), cellular components (CC), and molecular functions (MF). We conducted a DAVID enrichment analysis of the 157 DEPs from the GO and KEGG analyses. The top ten terms in each category were sorted based on the enrichment score, and a histogram was generated (Fig. 5C). The highest-ranking terms in BP included positive regulation of receptor-mediated endocytosis, adipose tissue development, 2-oxoglutarate metabolic process, protein localization to the plasma membrane and mitochondrial electron transport, and ubiquinol to cytochrome c (Figs. 5C, 6A). Enrichment in CC was observed in the cytoplasm, dendrites, and perinuclear regions of the cytoplasm (Fig. 5C), while enrichment in MF involved protein binding, transaminase activity, mRNA binding, and ganglioside GM1 binding (Fig. 5C). KEGG pathway enrichment analysis of the DEPs, conducted using Fisher exact test, revealed their involvement in crucial pathways. These pathways included metabolic pathways, the phosphatidylinositol signaling system, sphingolipid metabolism, thermogenesis, the HIF-1 signaling pathway, and the insulin signaling pathway (Fig. 6C). Most of these pathways are closely related to metabolism.

Figure 5 Description of DEPs between the control group and the model group.

(A) A volcano plot depicting the DEPs between the Control group and the Model group when FC > 1.2. Grey represents nondifferentiated proteins, upregulated DEPs are represented in orange, and downregulated DEPs are represented in blue. (B) Subcellular localization of DEPs between Control group vs. Model group. (C) GO function analysis histogram between Control group vs. Model group. BP is marked by dark cyan; CC is marked by sienna and MF is marked by steel blue. (N = 3 per group).

Figure 6 Enrichment analysis of DEPs.

(A) Annotation term levels of DEPs between Control group vs. Model group are indicated as BPs; (B) annotation term levels of DEPs between Model group vs. Treat group are indicated as BPs; (C) bubble chart of the enriched KEGG pathways between Control group vs. Model group; (D) bubble chart of the enriched KEGG pathways between Model group vs. Treat group. (N = 3 per group).

Identification of differential proteins associated with PCSK9 inhibitors

In order to further explore which proteins are targeted by PCSK9 inhibitors, we analyzed the DEPs between the Treat group and the Control group. The volcano plot displayed the differential proteins, with 40 upregulated and 56 downregulated DEPs between the Treat and Model groups (Fig. 7A). WoLF PSORT was employed to predict the subcellular localization of the DEPs, and the majority of the proteins were found to be located in the nucleus and cytoplasm (Fig. 7B). We conducted a DAVID enrichment analysis of the 96 DEPs for GO and KEGG. The top ten terms in each category were sorted based on enrichment score, and a histogram was generated using the results (Fig. 7C). The results indicated that the BP associated with these DEPs included negative regulation of mitochondrial membrane permeability involved in the apoptotic process, positive regulation of cell migration by the vascular endothelial growth factor signaling pathway, endosomal transport, positive regulation of the apoptotic process, and negative regulation of endopeptidase activity (Figs. 6B, 7C). The CC linked to these DEPs encompassed the mitochondria, synaptic vesicles, cytoplasmic vesicle membranes, dendrites, and cell bodies (Fig. 7C). The DEPs were also associated with specific MF terms, such as protein binding and actin binding (Fig. 7C). KEGG pathway enrichment analysis of the DEPs using Fisher exact test revealed their involvement in retrograde endocannabinoid signaling and the oxytocin signaling pathway (Fig. 6D).

Figure 7 Description of DEPs between the Model group and the Treat group.

(A) A volcano plot depicting the DEPs between the Model group and the Treat group when FC > 1.2. Grey represents nondifferentiated proteins, upregulated DEPs are represented in orange, and downregulated DEPs are represented in blue. (B) Subcellular localization of DEPs between Model group vs. Treat group. (C) GO function analysis histogram between Model group vs. Treat group. BP is marked by dark cyan; CC is marked by sienna and MF is marked by steel blue. (N = 3 per group).

Validation of target proteins associated with PCSK9 inhibitors

Among the Control and Model groups, a total of 157 DEPs were identified (59 upregulated and 98 downregulated DEPs). Between the Model and Treat groups, 96 DEPs were identified (40 upregulated and 56 downregulated). Notably, 13 overlapping DEPs were shared between the Control group / Model group and Model group / Treat group (Fig. 8A). PCSK9 inhibitor treatment exhibited a reverse regulatory effect on 12 of the 13 DEPs (Table 1). Among these 12 genes, special attention was given to transthyretin (Ttr), and we conducted further examination of the expression level of the Ttr protein using PRM and IHC (Figs. 8B–8D). The results obtained from PRM and IHC were consistent with the findings of the proteomic analysis. As anticipated, PRM and IHC demonstrated a decrease of Ttr in DACD, followed by an increase after intervention with PCSK9 inhibitors. Our data suggest that PCSK9 inhibitors may achieve therapeutic effects by modulating the expression of this critical protein.

Figure 8 Venn diagram and expression of transthyretin (Ttr).

(A) The Venn diagram indicated a comparison of DEPs between the Control group vs. Model group and Model group vs. Treat group; (B) The PRM results of the Ttr in rat hippocampus tissue (N = 3 per group); (C) Representative IHC of Ttr in hippocampus tissue from each group. Scale bars: 50 µm; (D) Quantification of Ttr staining. (N = 5 per group).

Table 1 Thirteen overlapping DEPs between the control group vs. Model group and Model group vs. Treat group.

Protein	Protein name	Gene name	Average model	Average control	Average treat	Model/Control	t test p value	Treat/Model	t test p value	
ENSRNOP00000078753	Rho/Rac guanine nucleotide exc	Arhgef18	199,470	139,893.3333	119,636.6667	1.425872093	0.020037804	0.599772731	0.001867489	
ENSRNOP00000078432	FHF complex subunit HOOK int	Fhip1b	279,593.3333	204,293.3333	209,426.6667	1.368587652	0.030075137	0.749040273	0.01185077	
ENSRNOP00000093734	kinase non-catalytic C-lobe do	Kndc1	319,123.3333	241,743.3333	260,210	1.320091557	0.045238348	0.815390079	0.017604052	
ENSRNOP00000085385	Uridine monophosphate synthetase	Umps	627,906.6667	481,710	500,840	1.303495187	0.001681136	0.797634468	0.018977367	
ENSRNOP00000073276	SPG7 matrix AAA peptidase subuni	Spg7	494,610	395,106.6667	401,346.6667	1.251839166	0.018604163	0.811440664	0.012400527	
ENSRNOP00000069996	Protein phosphatase 1, regul	Ppp1r8	40,107.66667	53,602	63,487	0.748249443	0.01456521	1.582914322	0.029277502	
ENSRNOP00000012612	Opioid growth factor receptor [	Ogfr	111,197.3333	153,783.3333	147,303.3333	0.723077923	0.007854231	1.324702031	0.038661823	
ENSRNOP00000097987	Ribosomal protein S6	Rps6	699,196.6667	973,313.3333	881,683.3333	0.718367501	0.007324072	1.260994761	0.001623219	
ENSRNOP00000069185	Phosphatidylinositol binding	Picalm	22,817.5	34,189.66667	33,449.66667	0.667380008	0.000592317	1.46596545	0.012255132	
ENSRNOP00000022113	Transthyretin	Ttr	517,893.3333	799,570	682,216.6667	0.647714813	0.005682764	1.317291849	0.00297421	
ENSRNOP00000064261	PDZ and LIM domain 4	Pdlim4	137,840	255,026.6667	212,386.6667	0.540492498	0.00133532	1.540820275	0.028163458	
ENSRNOP00000066155	Aminomethyltransferase	Amt	62,978.33333	156,666.6667	133,346.6667	0.401989362	0.020614971	2.117341943	0.006903454	
ENSRNOP00000087236	Thyroid hormone receptor intera	Trip4	71,012	96,818.6667	41,280	0.733454	0.035161	0.58131	0.034833	

Discussion

Increasing attention has been paid to DACD in recent years due to its detrimental impact on individuals with T2DM. DACD may progress to dementia, which can significantly diminish the quality of life of those affected. Currently, our understanding of the pathogenesis of DACD is limited. Unraveling the mechanisms underlying DACD and developing therapeutic strategies will be advantageous in preventing the progression of DACD to dementia and improving long-term outcomes for these patients. The study addresses an important gap in understanding DACD and the potential role of PCSK9 inhibitors, contributing valuable insights to the field.

PCSK9 can bind to LDLR and effect receptor degradation, leading to a reduction in the overall density of LDLR on the surface of hepatocytes (Grefhorst et al., 2008). PCSK9 not only plays a crucial role in lipid metabolism but also has a significant role in glucose homeostasis (Hachem et al., 2017). Previous evidence has indicated a significant increase in the PCSK9 concentration levels in patients with T2DM (Feng et al., 2023; Han et al., 2020; Walus-Miarka et al., 2021), suggesting that PCSK9 may have a certain influence on T2DM. The significance of PCSK9 in mediating T2DM has recently been confirmed (Macchi et al., 2021; Seidah & Prat, 2022). In the brain, PCSK9 functions in cholesterol homeostasis by regulating LDLR degradation and LDL uptake in neurons and astroglial cells, with implications for neurite outgrowth and cognition (Adorni et al., 2019). Furthermore, PCSK9 has been shown to promote neuronal apoptosis through activation of the Bcl-2/Bax/caspase-3 signaling pathway (Wu et al., 2014). Previous studies have indicated that PCSK9 disrupts brain cholesterol homeostasis, cellular apoptosis, BACE1 expression, and Aβ generation (Wu et al., 2014; Zhao et al., 2017), potentially leading to cognitive decline.

It was reported previously that monoclonal antibodies that target PCSK9 (such as evolocumab and alirocumab) can alleviate vascular events, including stroke (Castilla-Guerra, Fernandez-Moreno & Rico-Corral, 2019). PCSK9 inhibitor pretreatment has also been shown to prevent brain injury caused by myocardial ischemia/reperfusion damage by reducing dendritic spine loss, attenuating microglial overactivation, and inhibiting Aβ aggregation (Apaijai et al., 2019). In addition, research has indicated that PCSK9 inhibitors improve metabolic dysfunction, brain function impairment, and cognitive decline in insulin-resistant rats (Arunsak et al., 2020). Thus, PCSK9 inhibitors have demonstrated beneficial effects on brain function in stroke, myocardial ischemia/reperfusion injury, and insulin resistance-related cognitive impairment models. It has been demonstrated that PCSK9 inhibitors themselves do not impair cognitive function (Seijas-Amigo et al., 2023). Furthermore, we chose these dosages because previous studies have demonstrated their beneficial effects on lipid abnormalities, obesity, and peripheral insulin resistance in rats (Thonusin et al., 2019).

The MWM test is one of the most commonly used methods for assessing learning and memory abilities. In this study, both the Model group and the Treat group exhibited longer escape latencies compared to the Control group, and spent less time in the target quadrant and crossed the platform fewer times. These findings suggest a negative impact of T2DM on the learning and memory abilities of rats. Following treatment with PCSK9 inhibitors, not only was the escape latency reduced in the Treat group compared to the Model group, but the Treat group also spent more time in the target quadrant and performed more platform crossings relative to the Model group. These results indicate that PCSK9 inhibitors improve the learning and memory abilities of rats, possibly by promoting functional recovery of the impaired neural system.

Inflammatory cytokines such as IL-6, IL-1β, and TNF-α are renowned for their role in neuronal death, as these pro-inflammatory cytokines can induce neuroinflammation, subsequently leading to oxidative stress through the induction of ROS production and reduction of antioxidant levels (e.g., SOD and GSH) (Wang et al., 2020). Experimental studies have shown that diabetic patients release pro-inflammatory cytokines such as IL-1β, IL-6, and TNF-α, which may lead to neuronal death and accelerate neurodegenerative changes in Alzheimer’s disease (AD) (Puig et al., 2012). Marioni et al. (2010) confirmed the association between inflammation and cognitive impairment in diabetic patients. Gorska-Ciebiada et al. (2015) found that cognitive dysfunction in diabetic patients was associated with higher levels of inflammatory markers (CRP, IL-6, and TNF-α). Additionally, in the hippocampus of DACD mice, there was an elevation in inflammatory cytokines IL-1β, IL-6, and TNF-α (Nan et al., 2022). These studies indicate the significant role of inflammatory factors in DACD. Our research found that PCSK9 inhibitors can reduce the expression of IL-1β, IL-6, and TNF-α in hippocampal tissue.

Cholesterol in the brain accounts for more than 20% of the total cholesterol in the human body (Hussain et al., 2019). Cholesterol plays a necessary role in the formation and maintenance of synapses, and the steady-state of serum cholesterol reflects to some extent the cholesterol balance of specific organs (Zhang et al., 2021). Lipid parameters including TC, LDL-C, and TG are associated with cognitive impairment in various diseases (Ma et al., 2024). Low levels of LDL may be related to cognitive dysfunction (Beydoun et al., 2011; Evans & Golomb, 2009; Muldoon et al., 2000, 2004). Given that PCSK9 inhibitors can induce extremely low LDL-C levels, concerns are now more urgent. Recently, several large clinical trials have found no association between extremely low LDL-C levels and cognition (Gencer et al., 2020; Giugliano et al., 2017a, 2017b; Robinson et al., 2017). In contrast, one study suggested that low LDL-C levels may reduce the risk of AD (Benn et al., 2017). A recent study showed that low LDL levels were associated with a slower decline in cognitive function within 2 years (Hua et al., 2021). In this study, PCSK9 inhibitors reduced LDL and TC levels in the serum of T2DM rats, and improved cognitive function in the rats.

The hippocampus is a crucial brain region that has a key role in learning and memory (Rocca et al., 2018). In this study, PCSK9 inhibitor treatment improved the damaged neuronal morphology in the hippocampal CA1 region. Additionally, TEM observations indicated that PCSK9 inhibitor treatment protected against cognitive impairment in T2DM rats by reducing neuronal damage. These findings suggest that PCSK9 inhibitors demonstrate potential for improving cognitive deficits in T2DM.

4D label-free quantitative proteomics is a high-throughput proteomics technique that allows for faster, more sensitive, and more accurate identification and quantification of a wide range of proteins (Meier et al., 2018). In this study, a 4D label-free quantitative proteomics approach was employed, in conjunction with bioinformatics analysis, to delineate the proteomic landscape of DACD rats treated with PCSK9 inhibitors and to investigate the potential molecular mechanisms underlying the action of PCSK9 inhibitors on DACD. In this study, we identified 5,307 quantifiable proteins in the rat hippocampal tissue samples. The results revealed that, compared to the Control group, the Model group had 59 upregulated DEPs and 98 downregulated DEPs, while the Treat group had 40 upregulated DEPs and 56 downregulated DEPs compared to the Model group. GO is an internationally standardized gene annotation classification system that provides a comprehensive understanding of molecular functions, including BPs, CCs, and MFs (Thomas, Mi & Lewis, 2007). Pathway analysis is the most widely used method in bioinformatics, and it can help us to comprehensively and systematically understand the cellular biological processes involved in disease mechanisms (Xu et al., 2020). Our bioinformatics analyses of the DEPs collectively indicated that the DEPs in the Control group vs. Model group were primarily enriched in metabolic pathways, phosphatidylinositol signaling system, sphingolipid metabolism, thermogenesis, the HIF-1 signaling pathway, and the insulin signaling pathway. These pathways are primarily related to metabolism, indicating that metabolic pathways play a significant role in the pathogenesis of DACD. In contrast, the DEPs in the Model group vs. Treat group were mainly enriched in the retrograde endocannabinoid signaling and oxytocin signaling pathway. This result suggested that, in DACD, the retrograde endocannabinoid signaling and oxytocin signaling pathways may be involved in the neuroprotective effects of PCSK9 inhibitors. Retrograde endocannabinoid signaling is a widespread endogenous signaling system that not only participates in maintaining the body’s homeostasis but also regulates neuronal and glial activities. It has the ability to precisely control the timing and spatial processes of adult hippocampal neurogenesis. The regulation of neurogenesis by retrograde endocannabinoid signaling presents potential for treating emotional and memory disorders associated with neurobehavioral and neurodegenerative diseases (Oddi, Fiorenza & Maccarrone, 2023). Research has also revealed that the inhibition of retrograde endocannabinoid signaling could serve as a therapeutic strategy for neurodegenerative diseases (Chen, 2023). Oxytocin, a small peptide composed of nine amino acids, exhibits neuroprotective effects in numerous neurological disorders and is closely associated with the oxytocin signaling pathway and neurodevelopmental disorders (Ripamonti et al., 2017).

Among the identified 13 overlapping DEPs in the Control group vs. Model group and Model group vs. Treat group, 12 were proteins with reverse regulation upon PCSK9 inhibitor treatment, making them potential candidate targets for explaining the hippocampal protective effects of PCSK9 inhibitors. Among these 12 genes, particular attention was given to Ttr, and we further assessed the expression level of the Ttr protein using PRM and IHC. A previous study revealed that Ttr has neuroprotective effects in Alzheimer’s disease (AD) and cerebral ischemia in the central nervous system. However, its role in DACD has not yet been reported. Therefore, in this study, we selected Ttr for proteomic validation. The PRM and IHC results were consistent with the findings of the proteomic analysis: the Ttr levels were decreased in DACD, and they increased following intervention with PCSK9 inhibitors. Our data suggest that PCSK9 inhibitors may achieve their therapeutic effects by modulating the expression of this key protein. Ttr is a secreted protein primarily synthesized by the liver and choroid plexus of the brain (Pagnin et al., 2021). In the brain, as well as being expressed in the choroid plexus epithelial cells, Ttr is expressed in the neurons and oligodendrocytes (Gomes et al., 2018; Li et al., 2011; Zhou et al., 2019). In addition to its role as a transporter of thyroid hormone in the brain, Ttr has also been demonstrated to have other functions, including neuroprotective effects in hippocampal neurons and pathological conditions such as brain ischemia (Gomes et al., 2016; Santos et al., 2010) and AD (Buxbaum et al., 2008; Oliveira et al., 2011; Silva et al., 2017). In an AD mouse Model, a reduction in Ttr levels was associated with an increase in pathological changes (Aleshire et al., 1983; Soprano et al., 1985). In AD patients, the levels of Ttr in the cerebrospinal fluid (Serot et al., 1997) and plasma (Ribeiro et al., 2012) were found to be reduced. In cerebral ischemia, the levels of Ttr in the cerebrospinal fluid were found to influence the survival of vulnerable neurons (Santos et al., 2010). The first description of reduced cerebrospinal fluid Ttr levels in AD patients dates back to 1986 (Elovaara, Maury & Palo, 1986). Recently, studies have shown the significant decrease in the cerebrospinal fluid Ttr levels in patients with AD (Vieira & Saraiva, 2014). Furthermore, decreased levels of Ttr have been observed in the cerebrospinal fluid of patients with schizophrenia and amyotrophic lateral sclerosis (ALS), as well as in degenerating motor neurons in ALS (Ranganathan et al., 2005). All these observations suggest that Ttr serves as a neuroprotectant in the central nervous system.

Under conditions of excitotoxicity in vitro and in a mouse model of cerebral ischemia, Ttr has been identified as a determinant of hippocampal neuron survival, neurite growth, and preservation (Gomes et al., 2016). Ttr has been shown to enhance neurite survival and growth in hippocampal neurons by interacting with the receptor megalin and activating extracellular signal-regulated kinases, protein kinase B, and tyrosine protein kinase Src, ultimately leading to upregulation of the cAMP-response element binding protein transcription factor and favorable expression of antiapoptotic Bcl2 protein family members (Gomes et al., 2016). Ttr has also been demonstrated to elevate intracellular calcium levels through NMDA receptors in a Src/megalin-dependent manner, which can account for the capacity of Ttr to promote neurite growth (Gomes et al., 2016). In addition to interacting with megalin, Ttr has been demonstrated to upregulate the expression of this receptor (Gomes et al., 2020). Ttr therefore plays a crucial role in neuronal plasticity, synaptic activity, neurite growth, and neuronal activity (Magalhaes, Eira & Liz, 2021). Interestingly, in Ttr knockout mice, the absence of Ttr accelerates age-related memory impairment (Sousa et al., 2007), and reduced Ttr expression is observed in rats with age-related memory deficits (Brouillette & Quirion, 2008). The hippocampus is known to have a crucial function in the process of memory formation, and a lack of Ttr has been linked to deficits in memory. Consistent with the above research findings, our study revealed a decrease in Ttr expression in the hippocampus of DACD rats. Additionally, we found that PCSK9 inhibitors could increase Ttr expression in DACD rats. However, further research is needed to provide more evidence to elucidate the role and mechanisms of PCSK9 inhibitors in the cognitive function related to DACD. Furthermore, some studies have identified Ttr as a potential plasma biomarker that can aid in the diagnosis, reflect the severity, and assess the progression of AD. As neurodegenerative diseases, AD and DACD share many similarities, and further research is needed to determine whether Ttr can also be identified as a candidate plasma biomarker for DACD.

Our study has several significant strengths. Firstly, to the best of our knowledge, our study provides the first experimental evidence that PCSK9 inhibitors can improve cognitive dysfunction in T2DM rats. Secondly, our research is the first to discover a potential association between cognitive dysfunction in T2DM rats and decreased expression of Ttr in hippocampal tissue. Thirdly, our study is the first to suggest that the improvement of cognitive dysfunction in T2DM rats by PCSK9 inhibitors may be associated with increased expression of Ttr in hippocampal tissue. This study had some limitations. First, the small sample size may limit the generalizability of the findings. Future studies with a larger sample size could strengthen the validity of the results. Additionally, this study is in the preclinical stage, and currently, these findings cannot be extrapolated to humans. Further validation is needed in clinical trials in the future. Furthermore, the function of Ttr needs to be confirmed by laboratory and clinical data. Although our PRM analyses demonstrated the reliability of the predicted results, future studies should aim to confirm that the interaction between DEPs underlies the pathogenesis of DACD.

Conclusion

This study provides proteomic evidence for the use of PCSK9 inhibitors in the treatment of DACD. Our results suggest that PCSK9 inhibitors have a protective effect on cognitive function in DACD rats. These protective effects may be associated with the mitigation of hippocampal tissue inflammation, retrograde endocannabinoid signaling, and the oxytocin signaling pathway. Subsequent validation experiments indicated that Ttr may serve as a target of PCSK9 inhibitors, suggesting Ttr’s potential as a candidate protein for future therapeutic development against DACD. These promising findings suggest that PCSK9 inhibitors show potential as treatment for DACD.

Supplemental Information

Supplemental Information 1 Results of quantitative proteomics in the hippocampus of diabetes-associated cognitive dysfunction (DACD).

“Protein”, “Protein Name”, “Gene Name”, “ProteinGroup”, “FastaHeaders”, “Proteins”, “Peptides”, “RazorUniquePeptides”, “UniquePeptides”, “Coverage”, “MolWeight”, all are used to describe proteins. “t test p value” represents the protein differential expression between the Model group and Treat group, P < 0.05 significant. “Treat/Model” represents fold change of the protein differential expression between the Model group and Treat group. “red” means up-regulated protein expression, and “green” means down-regulated protein expression. Columns L to S respectively represent the relative expression of the differentially expressed peptides detected in three replicates of mass spectrometry in DACD rats and normal controls.

Supplemental Information 2 Results of PRM in the hippocampus of diabetes-associated cognitive dysfunction (DACD).

“Protein Name” and “Peptide Sequence” are used to describe proteins. Columns C to K respectively represent the relative expression of protein detected in three replicates of proteomics in DACD rats and normal controls. Columns D to L respectively represent the relative expression of peptides detected in three replicates of proteomics in DACD rats and normal controls.

Supplemental Information 3 qRT-PCR, Immunohistochemistry, Western blot weight, and glycolipid metabolism data.

Supplemental Information 4 The ARRIVE guidelines 2.0: author checklist.

Supplemental Information 5 Western blot band image: PCSK9, LDLR, and β-actin.

Additional Information and Declarations

Competing Interests

Author Contributions

Animal Ethics

Data Availability

The authors declare that they have no competing interests.

Yang Yang conceived and designed the experiments, performed the experiments, analyzed the data, prepared figures and/or tables, and approved the final draft.

Yeying Wang conceived and designed the experiments, analyzed the data, prepared figures and/or tables, and approved the final draft.

Yuwen Wang performed the experiments, prepared figures and/or tables, and approved the final draft.

Tingyu Ke conceived and designed the experiments, authored or reviewed drafts of the article, and approved the final draft.

Ling Zhao analyzed the data, prepared figures and/or tables, authored or reviewed drafts of the article, and approved the final draft.

The following information was supplied relating to ethical approvals (i.e., approving body and any reference numbers):

The Research and Animal Ethics Committee of Kunming Medical University (Kunming, China) approved the study (Approval number: kmmu20211142).

The following information was supplied regarding data availability:

Raw data are available in the Supplemental Files.

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
