# Peer review of "PCSK9 inhibitor effectively alleviated cognitive dysfunction in a type 2 diabetes mellitus rat model"

_PeerJ, doi:10.7717/peerj.17676_

## Round 0.1 · original submission · Major Revisions

While all three reviewers express some positive views, all raise substantive concerns which will require significantly more experimental work.

In no particular order:
I have substantial concerns due to the absence of any quantification to support the claims made regarding hippocampal pathology. Some suggestions from reviewer-1 are deemed essential on this point.

I would like to see the individual animal data displayed on the same graphs. I also think that is is difficult to draw conclusions since the control group did not receive the treatment as well. This is a major limitation.

Please carefully address the paradoxical aspect of the study's findings. It would be beneficial to expand the discussion section to thoroughly explain and reference the paradox where PCSK9 inhibitors prevent cognitive decline, despite existing literature often linking very low levels of LDL-c with potential cognitive function impairment. This exploration could include examining whether this relationship differs in a subgroup of patients with type 2 diabetes and discussing any relevant previous studies on this topic. Clarifying this paradox will provide a more comprehensive understanding of the study's findings and their implications in the broader context of current research.

Carefully attention to all of these points, plus those mentioned by the reviewers is required. Address all these in your revised paper together with a detailed letter explaining any changes you have made.

Reviewer 1 ·

Basic reporting

The paper is clearly written, with sufficient context.
The figures are nicely presented; however, I would suggest that the panels in Figure 1 should be rearranged such that Figure 1C becomes 1B as it relates to the quantification of 1A and this would be more logical.
Furthermore, all data displayed in Figure 1 panels C-D should be displayed so that the reader can see a symbol for the data for individual rats. It is impossible to see if the data is normally distributed in the way that its is currently presented.

Experimental design

The authors should clarify why only male rats were used for the MWM, and how sample size of 8 was calculated.

A control + treatment group is not included here. Therefore, any differences demonstrated with the PCSK9 inhibitor could be independent from the Model/disease?

Validity of the findings

In Figure 1A, the representative plot looks like the rat is displaying thigmotaxis behaviour (circling around the edge of the arena). Can the authors comment if this is reflective of the entire group, and if so, this may be suggestive of an anxiety phenotype, and this may confound the results since the animals may be too anxious to move into the centre to find the platform. Can the authors comment on this?
On lines 214-217, the wording is very confusing. Suggest reword to make the conclusions clearer.

The example traces provided in Figure 1B do not look any different between the control vs. Model vs. Treated animals, and do not seem to support the data in Figure 1D-E. The individual animal data (should be displayed on the graph) would be interesting to see here.

I think it would be very valuable to use in parallel a second measure of cognitive function with the same cohort of animals, but appreciate that this may not be possible here without repeating the entire in vivo study. A second readout would substantiate the effects on cognitive function.

In Figure 2A-D there is no attempt to quantify the data. It is therefore impossible to make the claims in the text regarding effects of the HFD on neuronal disarray. Similarly, the authors cannot say that the treatment "significantly ameliorated pathological alterations" as without quantification, no statistical analyses have been conducted. The authors should quantify the data from Figure 2A-D to support their claims. It appears to me that the images for Control animals could be taken from a distinct part of the hippocampus from the Model image. To help with this, the entire hippocampus should be shown, with a zoomed in image of the relevant region. The authors also make a claim regarding "reduced neuronal counts". How have the neurons been counted? Usually one would count NeuN -positive cells using immunohistochemistry and provide quantification of these cells within a given area. The legend for this figure does not make clear if all N=8 from each treatment group have been analysed. I do not think that the claims related to Figure 2A-D are supported by the data in its current form and without quantification.

Reviewer 2 ·

Basic reporting

General Comments
• The article is well-structured, with a clear hypothesis, relevant methods, significant results, and a comprehensive discussion. It aligns with professional scientific writing standards and contributes valuable information to the field.
Clear and Unambiguous, Professional English Used Throughout
• Assessment: The article is written in clear, unambiguous, and technically correct English. It adheres to professional standards of expression and courtesy.
• Suggestion for Improvement: No significant changes are required in this area.
Literature References, Sufficient Field Background/Context Provided
• Assessment: The article provides a sufficient introduction and background, placing the research within the broader field of diabetes-associated cognitive dysfunction (DACD) and PCSK9 inhibitors. It references relevant prior literature effectively.
• Suggestion for Improvement: Ensure that all key studies in the field are referenced, particularly recent advancements that might have a direct impact on the study's findings. I would suggest to include last research in this field , overall in Real World settings and clinical field. You reference EBBINGHAUS study from ANGEM but I would suggest to reference more recent articles like "Cognitive Function with PCSK9 Inhibitors: A 24-Month Follow-Up Observational Prospective Study in the Real World-MEMOGAL Study." (PMID: 37612529), current references are necessary.
Professional Article Structure, Figures, Tables. Raw Data Shared
Raw Data: The article does not explicitly mention if all raw data have been made available, as per the Data Sharing policy.
Suggestion for Improvement: Explicitly state the availability of raw data, or provide it in a suitable format in supplementary materials or a recognized data repository.

Experimental design

Assessment:
Well-Defined Research Question: The research question, exploring the impact of PCSK9 inhibitors on DACD, is well-defined and clear.
Relevance and Meaningfulness: The study addresses a significant and timely issue in the field of diabetes complications and cognitive dysfunction.
Knowledge Gap: The manuscript articulates how this research fills an identified knowledge gap, particularly regarding the lack of studies on the effect of PCSK9 inhibitors on DACD.
Suggestion for Improvement: Further emphasize the novelty of the research by highlighting the unique aspects of the study compared to existing literature.
Strengths:
1. Comprehensive Methodology: The study employs a robust experimental setup using a T2DM rat model, employing various sophisticated techniques such as Morris water maze test, histological analysis, TEM, and 4D label-free quantitative proteomics.
2. Clear Presentation of Results: Results are well-articulated, showing the effect of PCSK9 inhibitors on cognitive function in T2DM rats. The identification of potential target proteins and the reversal of transthyretin downregulation is a significant finding.
3. Relevance to Field: The study addresses an important gap in understanding DACD and the potential role of PCSK9 inhibitors, contributing valuable insights to the field.
Areas for Improvement:
1. Sample Size: The small sample size may limit the generalizability of the findings. Future studies with a larger sample size could strengthen the validity of the results.
I would include this point , the small sample size, in the limitations section clarifying the preclinical phase of this study.

Validity of the findings

Rationale and Benefit: The article could benefit from a clearer statement on how this research builds upon or differs from existing studies, particularly in terms of methodology or findings.
Suggestion for Improvement: Explicitly state how the study's methodology or findings differ from previous research in the field and the added value it brings to the existing literature.
Conclusions: The conclusions are well-stated and directly linked to the original research question about the role of PCSK9 inhibitors in DACD.
Support by Results: The conclusions are appropriately limited to the findings presented in the study.
Suggestion for Improvement:
Explain and thoroughly reference the paradox that could account for these results where PCSK9 inhibitors prevent cognitive decline in patients treated with these drugs, despite literature traditionally associating very low levels of LDL-c with potential cognitive function impairment. Could it be possible that in a subgroup of patients with type 2 diabetes, this relationship is different? What is known about this in previous studies, if any? Expand on this aspect in the discussion section of the study.

Reviewer 3 ·

Basic reporting

In this paper the authors evaluated the effect of a treatment with the PCSK9 inhibitor SBC-115076 on cognitive functions in a T2DM rat model (high fat diet + streptozocin). The main fiindings of the manuscript is that the pharmacological treatment ameliorated memory perfomances, evaluated by the Morris Water Maze test. This amelioration was pallaled by an improvement of hyppocampal alterations induced by DM. Finally by the mean of proteomic analyses, is was found that PCSK9 inhibition counteracted the downregulation of the hyppocampal protein tranthryretin. The authors conclude that the pharmacological inhibition of PCSK9 may be a potentially valuable strategy to treat diabetes-associated cognitive disfunction.

The topic is interesting, the English style is correct. The background provided is sufficient, although some report on the beneficial effect of PCSK9 inhibitors on cognitive functions needs to be added in the introduction (for instance in Alzheimer's Disease). Figures and Tables are clear and well described. All the results presented are relevant to the hypothesis.
Also in the introduction: proteomic is not the unique analysis performed in this study, please also mention other measurements.

Experimental design

The research question is well defined, the investigation has been conducted rigorously, by respecting the ethical standards. The methods are explained in sufficient details.

Validity of the findings

The results are supporting the conclusions, although some additional evaluation would be useful to complete the observation.

Results:
1. It would be valuable to add the evaluation of metabolic parameters in the three examined groups, i.e., body weight, fasting glucose, insulin levels etc. Does the treatment also ameliorate these DM-related parameters?
2. Figure 2. Quantification of the staining would be very helpful to make comparisons among groups.
3. Protein expression of transthryretin would be useful to be confirmed by western blot analysis.
4. It would be interestingly to add immunohistochemical analysis to evaluate the potential impact of PCSK9 inhibition on neuroinflammation-related parameters (see for example Brain Behav Immun. 2023 Nov 13;115:517-534).

Discussion:
How did you select the PCSK9 inhibitor (SBC instead of the monoclonal antibodies?).
Please speculate on a possible central effect of the PCSK9 inhibitor, based on its specific mechanism of action (extracellular antagonist of the LDLr). In this regards it would be very important to evaluate changes in hyppocampal LDLr expression before and after treatment.
Does this molecule potentially can cross the BBB?

---

## Round 0.2 · Minor Revisions

Please attend to the few remaining issues of reviewer 3 and then we can proceed to acceptance.

Reviewer 1 ·

Basic reporting

The paper is clearly written, with sufficient context. The comments I made previously have been addressed.
There are a number of suggested studies that haven't been completed due to feasibility of doing so. Although these experiments would have been valuable, I do not feel that the absence of them should preclude publication of the remaining data.

Experimental design

The sample size justifications are still not appropriate. I appreciate the attempted justification from the authors, but appropriate sample size calculations should be conducted rather than reliance on what other groups have done.

Validity of the findings

The conclusions and impact are well stated and the data largely support the authors conclusions.

Additional comments

The authors have largely addressed most of my previous comments / concerns.

Reviewer 2 ·

Basic reporting

I am pleased to see that the researchers have taken my suggestions seriously and made the appropriate modifications to their manuscript. I agree with the changes made and find the adjustments satisfactory. The inclusion of recent literature enriches the context and support for the study, ensuring that the manuscript stays current with the latest advancements in the field of PCSK9 inhibitors and their relationship with cognitive function. Additionally, the explicit statement regarding the availability of raw data reflects a commitment to transparency and reproducibility in research. Overall, these changes strengthen the quality and relevance of the study for the scientific community and field professionals.

Experimental design

The adjustments made in response to the concerns and suggestions about experimental design are thoughtful and effectively address the points raised. Incorporating the discussion about the study's sample size in the limitations section is a prudent choice, as it acknowledges potential constraints on the findings' generalizability. This addition not only enhances the manuscript's transparency but also underlines the importance of further research to corroborate these preliminary findings. By highlighting the study's preclinical nature and the necessity for future clinical trials, the authors clearly set the stage for the next steps in this research area. Additionally, the mention of needing to validate the function of Ttr with laboratory and clinical data further emphasizes the meticulous approach required for translating preclinical findings to human applications. Overall, I find these modifications to be well-reasoned and a valuable contribution to the manuscript, ensuring readers are well-informed about the context and limitations of the study.

Validity of the findings

The response from the researchers effectively highlights the novelty and significance of their findings, particularly emphasizing the unique contributions of their study to the understanding of PCSK9 inhibitors and their effect on cognitive dysfunction in T2DM rats. By clearly distinguishing their work from existing literature and addressing the paradox between PCSK9 inhibitors and LDL-c levels, they have strengthened the manuscript's validity. These adjustments not only clarify the study's unique position within the field but also enhance its relevance and potential impact.

Reviewer 3 ·

Basic reporting

The authors properly adresses all comments raised and the manuscript in my opinion has significantly improved.
I have few additional minor comments:

Abstract: lines 24-27, improve the sentences (nws it seems like a list).
Please specify that SBC-115076 has been used even in the abstract.
Figure 3A: control and models bars are significant different between each other? Please verify.
Figure 4D. Why an extracellular LDLr antagonist as SBC-115076, would reduce PCSK9 expression?
The discussion is a bit too long, please synthetize it.

Experimental design

no comments

Validity of the findings

ok

Additional comments

no comments

---

## Round 0.3 · accepted · Accept

Thank you for carefully addressing the remaining points.